# Nitric Oxide Function and Nitric Oxide Synthase Evolution in Aquatic Chordates

**DOI:** 10.3390/ijms241311182

**Published:** 2023-07-06

**Authors:** Annamaria Locascio, Giovanni Annona, Filomena Caccavale, Salvatore D’Aniello, Claudio Agnisola, Anna Palumbo

**Affiliations:** 1Department of Biology and Evolution of Marine Organisms, Stazione Zoologica Anton Dohrn, Villa Comunale, 80121 Naples, Italy; annamaria.locascio@szn.it (A.L.); giovanni.annona@szn.it (G.A.); filomena.caccavale@szn.it (F.C.); salvatore.daniello@szn.it (S.D.); 2Department of Research Infrastructure for Marine Biological Resources (RIMAR), Stazione Zoologica Anton Dohrn, Villa Comunale, 80121 Naples, Italy; 3Department of Biology, University of Naples Federico II, Via Cinthia 4, 80126 Naples, Italy; claudio.agnisola@unina.it

**Keywords:** embryonic development, nervous system, metamorphosis, invertebrate chordates, fish, amphibian

## Abstract

Nitric oxide (NO) is a key signaling molecule in almost all organisms and is active in a variety of physiological and pathological processes. Our understanding of the peculiarities and functions of this simple gas has increased considerably by extending studies to non-mammal vertebrates and invertebrates. In this review, we report the nitric oxide synthase (*Nos*) genes so far characterized in chordates and provide an extensive, detailed, and comparative analysis of the function of NO in the aquatic chordates tunicates, cephalochordates, teleost fishes, and amphibians. This comprehensive set of data adds new elements to our understanding of Nos evolution, from the single gene commonly found in invertebrates to the three genes present in vertebrates.

## 1. Introduction

Thanks to its peculiar features, nitric oxide (NO) has attracted a good deal of interest in recent decades. NO is a short-lived, fast-diffusing signal molecule acting in almost all organisms, including bacteria, plants, fungi, and metazoa [1,2,3]. Dissolved NO diffuses readily across plasmatic membrane and interacts with DNA, lipids, and proteins of neighboring cells [2].

NO importance in human physiology and pathology, recognized by the Nobel Prize award in 1998, has increased significantly through the discovery that this simple gas is intimately involved in regulating all aspects of human lives. Nowadays, it is clear that NO is involved in a large variety of vertebrates’ physiological processes, including proliferation, differentiation, apoptosis, macrophage activity, neurotransmission, cardiovascular homeostasis, and several pathological conditions, such as stroke, diabetes, and cancer [4]. Parallel studies performed in invertebrate animals have widened our knowledge of the various roles played by NO in different species (for a review, see [5]), revealing the ubiquitous occurrence and biological functions of this compound [3].

Unlikely, different methodologies with different levels of reliability are used to detect Nos in the different organisms so far investigated. Immunochemistry assays that do not have species-specific antibodies or an indirect NADPH-diaphorase method are generally less specific than the detection of mRNA. Consequently, in some cases, the results may not be congruent and therefore cannot always be generalized.

The majority of chordates, a monophyletic clade [6,7], live in aquatic environments, ranging from oceans to inland waters to estuaries, and are therefore adapted to a wide range of physicochemical conditions [8]. In this review, we report the function of NO in aquatic chordates, including tunicates, cephalochordates, teleost fishes, and amphibians, and highlight the conserved NO roles. This review mainly deals with the Nos-dependent NO production and function as a major component of NO homeostasis, also providing information on *Nos* genes found in chordates to obtain further insights into Nos evolution.

## 2. NO Homeostasis

NO is one of the main actors in the cellular redox landscape of living organisms, and its homeostasis is of paramount importance for its functional roles at all levels in an organism, from the molecular to the whole body (Figure 1). As a signaling molecule, NO typically exerts its physiological functions by (i) reversibly binding to hemoproteins, forming an iron-nitrosyl complex; (ii) S-nitrosylation of proteins (thiol residues); and (iii) reacting with amines (N-nitroso compounds) [9]. Thioldisulfide exchange reactions modulate protein structure and function and maintain cellular redox homeostasis [10]. The soluble guanylate cyclase (sGC), an almost ubiquitous hemoprotein, is a primary target of NO. The binding with NO via its ferrous heme activates by 100- to 200-fold the sGC enzyme [11]. Interestingly, among small diatomic signaling molecules, NO binds five-coordinate ferrous heme species, unlike O_2_ and CO, which bind to six-coordinate species [3]. NO can also bind to non-heme iron and copper ions [3]. The latter can be important in modulating cytochrome c oxidase and mitochondrial respiration [12,13,14].

NO needs to be produced in the proximity of the site of action being very reactive, and its diffusion gradient spans relatively short distances due to a series of scavenging reactions (Figure 1) [15]. In the presence of oxygen, NO may (i) rapidly react with water forming nitrite (NO_2_^−^); (ii) be inactivated to nitrate (NO_3_^−^) by oxygenated globins (as hemoglobin, myoglobin, and neuroglobin); (iii) generate peroxynitrite (ONOO^−^) or dinitrogen trioxide (N_2_O_3_), which in turn may form reactive nitrogen species and affect the cellular redox state. All reactive nitrogen species produced in these scavenging processes can contribute to protein, lipid, and DNA nitration [16], protein S-nitrosylation [9], and S-glutathionylation [17], thus affecting protein homeostasis [10,18].

Under normoxic conditions, NO is mainly produced by nitric oxide synthase (Nos), an enzyme that requires molecular oxygen and several cofactors, synthesizing NO and citrulline from L-arginine [19]. Citrulline can be recycled to L-arginine via the ornithine–urea cycle (OUC), a process mainly studied in vertebrates, although OUC enzymes appear to be widely present among prokaryotes and eukaryotes [20,21]. Nos can produce nanomolar to micromolar concentrations of NO, under constitutive or induced activity, respectively.

Under low oxygen environmental or tissue conditions, Nos-dependent NO production is insufficient, and alternative mechanisms are involved. In particular, globins can act as nitrite reductases, generating NO [15]. Indeed, NO_3_^−^ (via a putative nitrate reductase activity, as described, for example, in mammals and fish [22,23]) and NO_2_^−^ serve as a bioavailable reservoir of NO in blood and tissues (globin–NO cycling) [15]. NO can also be generated from NO_2_, NO_3_^−^, and S-nitrosothiols in a decomposition process mediated by UV light that has been proposed to modulate blood perfusion in the skin [24].

The regulation of NO production and scavenging is essential for the proper NO function as a signaling molecule. The NO profile in a cell or tissue is defined by its concentration, persistence, and spatial orientation, and is determined by both physical and chemical processes. Under normoxic conditions, the activity and rate of NO production are primarily determined by controlling Nos transcription and translation, and by post-transcriptional and post-translational modifications [25,26], as well as by Nos cofactors or substrate availability, while Nos-independent pathways are mainly regulated by changes in the oxygen tension [10]. Additionally, Nos activity can be modulated by hydrogen sulfide (H_2_S), a relatively poorly studied signaling molecule [27].

From the above, it is clear that the multitude of NO derivative species (NO_x_, see Table 1) produced in living cells have varying significance, depending on the diversity of the target reactions that these species can undergo in the complex biological environment (Figure 1) [27]. This complexity is at the base of the double-edged sword behavior of NO in biological systems. Its freely diffusing capacity, sufficiently long lifetime, and very high affinity with ferrous heme make NO a highly efficient signaling molecule. On the other hand, NO can be noxious as a source of strong oxidizing agents that can damage plasmatic membranes, proteins, and DNA [16].

The interaction with arginine homeostasis is critical in regulating NO production. The role of arginase in modulating Nos has been particularly investigated [29,30]. This interaction between Nos and arginase is more complex than simple substrate competition [31], depending on the complexity of cellular and whole-body arginine fluxes [30]. Arginine, which is semi-essential or conditionally essential in vertebrates, is the substrate for several enzymatic pathways leading to the biosynthesis of urea, ornithine, agmantine, creatine, citrulline, and glutamate [32]. Arginine is implicated in collagen synthesis and in the modulation of immuno-responses [33,34]. The complex interaction between arginine and NO homeostasis is epitomized by the so-called “arginine paradox”. In mammals, endothelial NO synthesis can be regulated by modulating extracellular arginine concentration, despite the fact that the intracellular arginine concentration usually exceeds the Km for arginine of eNOS. No definitive explanation for this paradox is yet available [33].

The overproduction of NO is at the base of a wide range of physio-pathological alterations, including neurodegenerative diseases, inflammation, and ischemia, mainly via peroxynitrite excess, protein S-nitrosylation, and nitration (SNOs) [35,36,37,38,39]

Apart from reduced or altered enzymatic expression or functionality, NO under-production can result from Nos uncoupling, in which the enzyme produces superoxide at the expense of NO, due to deficiency of the cofactor tetrahydrobiopterin or the substrate L-arginine, as well as Nos S-glutathionylation. Nos uncoupling strongly contributes to oxidative stress [40,41].

## 3. Nos Evolution in Deuterostomes

The *Nos* gene family has experienced loss and duplication events in several animal lineages, although the gene structure remains nearly unchanged [42].

In mammals, three *Nos* genes are reported, encoding for three distinct proteins often improperly defined as isoforms. Two of these genes, the neuronal (*nNos*) and the endothelial Nos (*eNos*), are constitutively expressed mainly in neurons and endothelium, respectively, whereas the third gene, the inducible Nos (*iNos*), is expressed in macrophages upon stressful stimuli. However, the different Nos genes are not exclusively expressed in a single cell type, e.g., neurons, endothelium, and macrophages, as initially believed. Moreover, the inducible Nos may act constitutively in some cells (see for details: [42,43]). Therefore, the above-mentioned classical nomenclature appears nowadays inadequate and, to avoid confusion, here we adopt the *Nos1*, *Nos2*, and *Nos3* gene nomenclature, which reflects the chronology of their identification (Table 2).

The corresponding proteins have different properties and localization. Synthetically, Nos1 is calcium-dependent and constitutively expressed in neurons and several other tissues, including muscles. Nos2 is calcium-independent and is inducible in macrophages but is also present in other tissues such as cardiac myocytes, glial cells, vascular smooth muscle cells, hepatocytes, chondrocytes, and keratinocytes. Nos3 is calcium-dependent and constitutively expressed in the endothelium but is also present in the myocardium, brain, and other tissues. These discrepancies are linked to structural differences: Nos1 and Nos3 possess an inhibitory loop sequence, responsible for their calcium-dependency, which is absent in Nos2; Nos1 has a PDZ domain, while Nos3 shows myristoylation and palmitoylation motifs [42,44].

The three distinct Nos present in deuterostomes probably originated from an ancestral Nos that structurally resembles the Nos1 for the presence of PDZ domain. Multiple Nos gene copies have been found in invertebrate deuterostomes, including sea urchin, Amphioxus, and early branching vertebrates like lamprey, resulting from lineage-specific gene duplication events (Table 3) [42,45]. On the contrary, a single Nos copy has been identified so far in the ascidian sea squirt (Table 3) [46].

Whole-genome duplication events (WGD) [56,57] and the consequent increase in genomes complexity have expanded the number of paralogs and their functional specialization in vertebrates: *Nos1*, *Nos2*, and *Nos3* (Figure 2). This gene repertoire is conserved in Sarcopterygians and early diverging Actinopterygians, as polypterids, acipenserids, bowfins, and gars, but shows a highly diversified landscape in teleosts. Teleosts underwent the teleost genome duplication (TGD) that produced, generally, an amplification of the gene repertoire (Figure 2) [58]. The loss of *Nos3* has been detected in all the analyzed teleosts, except for elephant fish, which still possess the complete Nos repertoire [54]. The evolutionary history of *Nos2* reflects a more complex scenario with multiple independent lineage-specific duplications, as found in cavefish, catfish, or northern pike, and 4R tetraploidization events occurred in cyprinids and salmonids (Cs4R and Ss4R, respectively) [54]. Despite all this, *Nos2* has been lost at the stem of Neoteleosts. The *Nos1* gene is instead ubiquitously present, as a single gene copy, in gnathostomes, except for duplicates deriving from 4R events [54].

It remains to be demonstrated if multiple *Nos* duplicates are involved in specific biological pathways (sub-functionalization) and whether a compensatory mechanism exists to functionally compensate the lost genes.

## 4. NO Function in Cephalochordates

In 2011, Andreakis and collaborators described three different *Nos* genes in the Amphioxus *Branchiostoma floridae*’s genome, called *NosA*, *NosB*, and *NosC*, which were not orthologues of the three human *Nos* genes based on phylogenetic analysis. The analysis of gene and protein structures allowed the identification of NosA and NosC as neuronal-like types, thanks to the presence of the N-terminal PDZ domain and of the inhibitory loop sequence. NosB misses both those domains, suggesting that it could represent an inducible-like Nos, similarly to the human Nos2 [42]. Nos genes have also been found in other cephalochordate species, such as *Branchiostoma lanceolatum*, *Branchiostoma belcheri,* and *Asymmetron lucayanum*, and thanks to a dedicated phylogenetic analysis using more cephalochordate species, it was confirmed that they originated from two cephalochordate-specific gene duplications [45]. The three *Nos* in Amphioxus show a complementary expression pattern, from early embryogenesis to adulthood, indicating a putative NO role in multiple processes spanning the entire Amphioxus’ life. NO deriving from NosB was hypothesized to have an important role during gastrulation, since *NosB* in *B. lanceolatum* was expressed in between endoderm and ectoderm at early and middle gastrula [45]. Later during embryogenesis, particularly during neurulation, NO from NosC was found to have a key role in the antero-posterior patterning of the pharyngeal region. For the normal development of the pharynx, NO cooperates with another important and ancient signaling molecule, Retinoic Acid (RA) [49]. During the successive developmental phases, both in *B. lanceolatum* and *B. floridae* larvae, Nos expression was observed in the intestine and the club-shaped gland [45,59]. The club-shaped gland is involved in the production of mucus for capturing food particles entering the mouth, and this suggests a role of NO in the feeding behavior that is a conserved feature in other animals [60,61,62,63,64,65,66]. The neuronal expression of *NosC*, detected from neurula to larva stages, is a strong indication of the NO’s Involvement in neurogenesis and neurotransmission [45,49]. *NosC* is the only Amphioxus *Nos* gene for which a putative enhancer, conserved among *B. lanceolatum*, *B. floridae*, and *B. belcheri*, has been identified using a transphyletic approach [52]. Such an enhancer was active in an in vivo reporter assay in the CNS of ascidian embryos and recapitulated the endogenous *Nos* expression, indicating conservation of the *Nos* regulatory machinery among invertebrate chordates [52]. The neuronal role of NO is important in the adult as well, where Nos and NO were identified in the cerebral vesicle and neurons of the neural cord [50,67]. In the adult *B. belcheri*, apart from a role in the nervous system, NO is implicated in the maturation of germinal cells, both egg and sperm, and in the immune response [50,51,67,68]. In particular, *Nos* transcripts were detected in macrophages in the lymphoid cavities of metapleural folds and of the branchial coelom as well as in the gut and the endostyle, which are known to be immune-system-associated organs [50,51,67]. At the time of the experiments, the above-mentioned authors ignored the existence of three different *Nos* genes in Amphioxus, and based on expression profiles shown in *B. lanceolatum*, it is plausible that they detected *NosA*, which is, in fact, exclusively expressed in adulthood [45].

## 5. NO Function in Tunicates

In tunicates, NO is involved in various processes from developmental to adult stages. *Nos* expression during the various phases of embryonic development has been analyzed by real-time PCR experiments in two ascidian species: *Ciona robusta* and *Herdmania momus*. Unfortunately, no data are present in the literature about Nos or NO localization in tunicate early stages of development, although a progressive increase in *Nos* gene expression is evident from neurulation onwards [46,69]. Most studies are centered on Nos expression and NO function during larval development and metamorphosis.

Metamorphosis represents a key developmental event in which there is a radical transition from a larval to a juvenile/adult body plan and the need to control two overlapping developmental programs simultaneously: larval and juvenile development [70]. All metamorphoses involve the differentiation of juvenile/adult structures, the degeneration of larval structures, metamorphic competence, and a habitat change [70]. In tunicates, these changes are entirely different in each of the classes. Whereas ascidians undergo a dramatic metamorphosis, members of the Appendicularia retain a larval body plan in the adult, and Thaliacea exhibit a direct development [71]. It has been shown that in *C. robusta*, NO exerts a fundamental role for the correct larval structural organization. Interestingly, dynamic expression patterns of Nos genes and proteins as well as NO signals in the nervous system, tail muscle, and epidermis seem to correlate with the different phases of metamorphosis induction and progression [5]. As larval development proceeds, a progressive shift of Nos expression and NO signal along the antero-posterior axis of the larva is observed that is intimately connected to the different phases of induction of larval settlement and tail regression. At the early and middle larva stages, Nos expression and NO localization in the anterior palps and CNS are necessary for settlement induction. Later on, at the late larva stage, Nos expression and NO signal along the tail nervous system, muscles, and notochord control change in the swimming behavior and induction of tail resorption [5,46].

Most of our knowledge of the involvement of NO in ascidian metamorphosis comes from the pharmacological treatments of larvae with NO donors or inhibitors of Nos or guanylyl cyclase. The modulation in NO levels in the ascidians *C. robusta*, *Boltenia villosa*, *Cnemidocarpa finmarkiensis*, and *H. momus* affect the frequency of metamorphosis, resulting in tail retraction and resorption that characterize the juvenile morphology [46,69,72,73,74]. These results are further supported by Nos gene and protein expression and NO localization in the tail epidermis and tail tip at all regression stages and NO diffusion in all tail tissues, including notochord and muscle cells in *C. robusta* [5,46] and *C. finmarkiensis* [73].

The detection of NOS, by the indirect histochemical method of NADPH-diaphorase activity, in specific regions of adult *Styela rustica* and *Molgula citrina* has suggested the involvement of NO in several functions, including in the processing of the sensory information, regulation of reproduction, immune response, regulation of respiratory and digestive functions, muscle and intestinal tones, and epithelial cell functioning, and the synthesis of secretory products [75].

Of particular interest is the involvement of NO in the immune function in some tunicates. Among the different types of blood cells of *Styela plicata*, the lymphocyte-like cells produce most of the NO, as revealed by NADPH-diaphorase and immunostaining with antibodies against inducible Nos [76]. Also, hemocytes of *Phallusia nigra* contain NO, detected using the cell-permeant 4-Amino-5-Methylamino-2′,7′-Difluorofluorescein Diacetate (DAF-FM-DA), which becomes fluorescent after reaction with NO [77]. No conclusive data were obtained by functional experiments in which NO production was investigated in hemocytes under inflammatory conditions after treatment with lipopolysaccharide (LPS) or zymosan A [76,77].

## 6. NO Function in Teleosts

Teleost species account for about half of all vertebrate species known today and have diversified into virtually every aquatic habitat on the planet. Salinity, temperature, and O_2_ availability have been the main drivers of fish evolution. The long evolutionary history and the wide variety of aquatic environmental conditions explain the high diversity in this group of animals.

As in mammals, in fish, NO is involved in a large number of physiological processes, including development, cardiovascular homeostasis, neurotransmission and neuromodulation, and immune defense. The signaling role of NO in fish has often been studied from the perspective of its involvement in animal responses and adaptation to environmental stress, i.e., any environmental condition or change that challenges the homeostatic and allostatic processes of animals. The ability of animals to cope with stress-dependent challenges represents a fundamental mechanism for maintaining organism homeostasis. NO is involved, for example, in the teleost response to osmotic stress [78], hypoxia [79], and thermal stress [80].

It should be noted that a large body of information on the role of NO in teleosts comes from a limited number of model species, mainly cyprinids and salmonids, the winner being the zebrafish embryo. This situation limits the possibility of thoroughly evaluating the functional role of NO in this group of organisms characterized by a large diversity. As reported in Figure 2, due to a complex evolutionary process, *Nos1* is present in all species of Gnatostomata so far examined, whereas *Nos3* is lost in Clupeocephala, and *Nos2* is duplicated in several species, but lost in Neoteleostei [54]. Therefore, apparently, only one gene, *Nos1*, accounts for all enzymatic NO synthesis and the wide range of NO roles in Neoteleostei.

### 6.1. Role of NO in Fish Reproduction and Development

As in mammals, in teleosts, NO is critical in both oocyte maturation and egg activation and maturation upon fertilization [15]. Nos plays a role in oocyte maturation [81], although a fine-tuning of NO appears necessary, as both too low and too high NO levels have negative consequences. The mechanism of such a fine-tuning process has yet to be clarified.

NO is involved in the calcium rise necessary for egg activation upon fertilization [82]. Interestingly, maternal zebrafish transcript reported no detection of the Nos enzymes in eggs and embryos until 6 h post fertilization [83]. Hence, it is unclear which source of NO is involved in egg activation. Globin X, which is expressed at high levels in the eggs at fertilization, has been proposed to play this role [15], given its efficient nitrite reductase activity [84].

The availability of fish larval models for studying the development of vertebrates, mainly zebrafish (*Danio rerio*) and medaka (*Oryzias latipes*), has also offered the opportunity to gain significant insights into the role played by NO in fish development.

In the developing zebrafish, the strongest NO-positive sites in 5-day post fertilization larvae are the notochord, bulbous arteriosus, and cranial bones [85]. Nos inhibitors did not completely abrogate the NO signal, suggesting that there are additional Nos-independent sources of NO in the developing embryo [85]. NO is involved in the control of vasculature during development [86,87,88] and stimulates angiogenesis and hematopoiesis [87,89]. The canonical NO signaling (NO activating sGC and catalyzing the synthesis of cGMP) targets several proteins, including Bone Morphogenetic Protein-4 (Bmp4), which is involved in determining the positioning of the heart during embryonic development [90].

NO plays a key role in the development and plasticity of the CNS during both embryonic and post-embryonic stages of life [91]. Studies performed in different species on the developing nervous system have implicated NO in neural differentiation, pathfinding, and synapse formation [91]. NO is involved in motor axon development and branching [92], and in the cranial neural crest patterning, differentiation and convergence during craniofacial morphogenesis [93]. It has been suggested that NO and/or the cGMP generated by soluble GC play a role in neuronal signaling and neuronal development in the medaka fish embryonic retina [94].

### 6.2. Role of NO in Teleost Cardiorespiratory Function, Osmoregulation, and Feeding

NO acts as a paracrine and endocrine modulator in the gills, heart, blood vessels, kidney, and intestine, playing a crucial role in highly integrated processes such as respiratory gas distribution, osmoregulation, and gut function.

Despite the absence of Nos3, there is plenty of evidence for a role of Nos-derived NO in the homeostasis of the cardiovascular system. Nos3 immunoreactivity (i.e., using anti-mammalian eNos, often indicated as eNos-like) has been found in the heart and endothelial cells of several species [53,95], and the role of NO in controlling cardiac function is well established [9,80,96]. The presence of Nos3-like and Nos2 in the different districts of the fish heart has been described (as reviewed in [97]), with evidence for an NO role as an autocrine and paracrine molecule modulating its functionality. Nos1 immunoreactive signals have also been detected in the goldfish heart’s intracardiac nervous system (ICNS) as part of the complex neuronal set necessary for fine-tuning cardiac function [98].

NO donors (e.g., SNP) and Nos inhibitors (e.g., L-NAME) affect vascular resistance, confirming its putative vasodilatory function in fish [86,99,100]. The occurrence of perivascular nitrergic neurons, with Nos1 as a likely primary source of NO, innervating the vasculature of some teleost species, is also described [53,101].

Few studies have also addressed the occurrence and role of the classical NO-sGC-cGMP pathway in the fish’s cardiorespiratory system. The occurrence of GC-activating protein (GCAP) genes has been demonstrated in pufferfish (*Fugu rubripes*), zebrafish, and medaka [90,102,103,104]. As in rats, mice, and humans, the genes encoding the sGCa1 and b1 isoforms are closely located in medaka, accounting for their transcriptional coordination [105]. Indirect evidence of sGC-dependent cardiac modulation by NO has been reported in adult goldfish (*Carassius auratus*) [106] and zebrafish larvae [107,108].

In the teleost gill vascular district, NO-dependent vasoconstriction, rather than vasodilation, has been described. For example, NO-cGMP signaling has been reported in the freshwater eel branchial vasculature [109], with NO-dependent vasoconstriction. Gill Nos1 immunoreactivity has been reported in killifish (*Fundulus heteroclitus*) [110], catfish *(Heteropneustes fossilis*) [111], *Gadus morhus* [112], African bonytongue (*Heterotis niloticus*), [113], and Antarctic fish (*Trematomus bernacchii* and *Chionodraco hamatus*) [114].

NO has been involved in controlling ion transport in gill and kidney epithelia in several species [110,115,116]. The NO-cGMP system is likely involved in the regulation of Na+/K+-ATPase (the leading actor of osmoregulation in fish) in the kidney of brown trout (*Salmo trutta*) [115]. Nos1 has also been localized in the rainbow trout kidney [117]. NO also plays a role in the regulation of NaCl transport in the intestines of marine and air-breathing fish [118,119]. Nos1, which is highly expressed in the middle intestine of rainbow trout (*Oncorhynchus mykiss*) [120], is upregulated during sea water acclimation [121].

The respiratory rhythm of zebrafish is affected by manipulating the NO system [122]. In larvae, NO donors increase ventilation, while the same manipulation inhibits adult ventilation [122]. NO and Nos have also been involved in the complex cardiorespiratory and osmoregulatory adjustments during aestivation in lungfish (*Protopterus sp*.) [123].

NO generally has an inhibitory effect on intestinal smooth muscle cells, and NOS has been found in gut neurons of several teleosts [120,124,125,126]. A nitrergic inhibitory tone has been shown in Atlantic cod and zebrafish intestine [127,128], but not in rainbow trout stomach [129]. The enteric neurons and nitrergic inhibitory tonus develop before the onset of exogenous feeding in zebrafish [128] and Atlantic cod [127]. The zebrafish mutant *lessen*, characterized by a reduced number of proliferating ENS precursor cells and a delayed and disturbed intestinal motility, presents a significantly lower number of nitrergic neurons than the wild type [130]. In *Anguilla anguilla*, parasitized by the worm *Contracaecum rudolphii*, Bosi et al. [131] discovered an increase in the density of nerve fibers harboring endogenous Nos1.

### 6.3. Role of NO in the Immune System of Fish

The teleost immune system comprises the most basic immune organs, immune cells, and molecules among vertebrates. The teleost innate immunity faces pathogen infection and initiates the adaptive immune response, which, compared with that in higher vertebrates, shows some limitations, such as a limited antibody library, slow proliferation, and maturation of lymphocytes and longer memory time [132].

Teleosts have monocytes/macrophages and neutrophils. These cells are considered professional phagocytes that recognize a wide range of particles or antigens and have a strong and efficient endocytosis ability [132]. Classically, activated macrophages express high levels of inducible NO synthase (Nos2). It has been reported that macrophages express Nos2 when stimulated by IFNγ, TNFα, or bacterial MAMPs [133]. Simultaneous production of superoxide and NO intermediates can form peroxynitrite, which additionally serves potent antiparasitic/antimicrobial functions [134,135].

The involvement of Nos2 in the response to infection has been identified and characterized in various teleost species. Fish macrophages’ diversity and plasticity are similar to that found in mammals [136]. Fish macrophages produce NO [137,138]. Nos2 can be detected at a high level when the pathogen *Enteromyxum scophthalmi* infects turbot [139]. Common carp macrophages stimulated by LPS or protozoan blood flagellate *Trypanoplasma borreli* produce a large amount of Nos2, as shown by RT-PCR [140]. Nos2 is also highly expressed in zebrafish, although not upregulated following *Frascinella* infection [141]. Nos2 expression in gills has also been reported in rainbow trout infected with Gram-positive pathogens [142]. These studies suggest that the production of NO by teleost macrophages can be a significant component of the teleost immune response to pathogens.

Mammalian macrophages not only play a role in initiating inflammation but also in the resolution phase of inflammation (macrophage polarization, with metabolically distinct pro-inflammatory, M1, and anti-inflammatory, M2, phenotypes) [136]. Macrophage polarization has also been reported in common carp *Cyprinus carpio carpio* [143,144], where the M1 macrophages are characterized by high NO production and low arginase activity, while the M2 macrophages are characterized by high arginase activity, so exploiting the Nos–arginase competition for substrates [144].

### 6.4. Role of NO in the Teleost Nervous System, and Control of Locomotion and Behavior

Nos-derived NO is an important neurotransmitter in the CNS that is involved in the regulation of most systemic functions, as well as learning, sensory processing, and other forms of neural plasticity [145].

In ray-finned fish, early expression of *Nos1* mRNA in the CNS has been shown [55,91,104,146]. In adult teleosts, peripheral organs contain Nos1, as in mammals [147,148]. The co-expression of Nos1 with serotonin and tyrosine hydroxylase-immunoreactive fibers in the skin and gills of the giant mudskipper, *Periophthalmodon schlosseri*, likely related to the amphibious lifestyle of this species, was found by Zaccone et al. [149].

Interestingly, NO signaling is involved in several aspects of fish behavior. Using zebrafish knockouts for the *Nos1* gene, Penglee et al. [150] reported high dopamine levels and a low inter-individual distance (high shoal cohesion) in shoaling juveniles of zebrafish related to the brain’s low levels of NO. Moreover, zebrafish Nos1 knockouts are less aggressive and display anxiety-like behavior [151]. Brain *Nos1* is among the genes associated with aggressiveness in female zebrafish [152]. Recently, fish oxytocin (also called isotocin) has been involved in the regulation of social fear contagion in zebrafish [153]. Since oxytocin can stimulate NO production [154], we can indirectly speculate on NO involvement in social fear in fish.

Although the distribution of Nos1-expressing cells seems to be conserved in most teleosts, with a high degree of detection in the telencephalon and diencephalon and reduced expression in more caudal brain regions [55,155], diversity of function can be observed. For example, in the African cichlid fish *Astatotilapia burtoni*, Nos1-expressing cells were found in the inner nuclear layer and from the olfactory bulbs to the hindbrain, including within specific nuclei involved in decision-making, sensory processing, neuroendocrine regulation, and the expression of social behaviors [156]. These authors demonstrated that NO signaling might play diverse roles depending on sex and male dominance, likely mediating the sensory perception required for structuring male social hierarchies or male courtship in this species characterized by territoriality, male hierarchy, and maternal mouthbrooding. Using RT-PCR, they reported the occurrence of Nos1-expressing cells in the retina [156], in agreement with the hypothesis of a significant role of NO signaling in modulating retinal light response [157], but not in the olfactory epithelia. Using immunocytochemistry, Nos1 has also been detected in the olfactory receptor neurons of tilapia, *Oreochromis mossambicus* [158] and in the retina of goldfish and catfish [159].

NO is among the major players involved in the neuromodulation of the locomotor central pattern generating (CPG) networks both in vertebrates [160] and invertebrates [161]. During animal development, the requirements of locomotor control systems may undergo significant changes at the molecular (receptors, ion channels, pumps, and transporters in individual neurons), cellular (changes in existing neurons or formation of new neurons), or systemic (new neuronal networks) levels in relation to the emergence of new or modified neuromuscular structures and behaviors [162]. In zebrafish, Nos1 mRNA expression is detected at 19 h post fertilization (hpf) in the forebrain, and Nos-positive neurons increase in the brainstem [91]. However, despite the fact that no evidence of NO directly modulating the swimming CPG has been reported so far, the zebrafish swimming system as a whole is profoundly modulated by NO signaling, which plays a role in regulating muscle innervation and neuromuscular transmission [92,163].

## 7. NO Function in Amphibia

In Amphibia, there is evidence of the role of NO as a signaling molecule in several functional processes. What is particularly studied is its role in Anura’s metamorphosis.

Nos immunoreactivity was found in the enteric nervous system of axolotl, *Ambystoma mexicanum* (Urodela) [164]. The NO/cGMP signaling pathway was functional, and its components were widely distributed throughout specific cell types in the outer and inner salamander retina [165].

*Xenopus tropicalis* possesses a *Nos3* gene, which is orthologous of mammalian *Nos3* (as determined using gene synteny and phylogenetic analyses). *Nos3* mRNA expression was highest in lung and skeletal muscle and lowest in the liver, gut, kidney, heart, and brain. Nos immunoreactivity was observed in the proximal tubule of the kidney and endocardium of the heart but not in the endothelium of blood vessels. Thus, *X. tropicalis* has a *Nos3* gene that appears not to be expressed in the vascular endothelium [166].

A putative role of Nos2 and Nos3 has been reported in the larval, pre-metamorphic, and adult phases of the newt *Triturus italicus* (Amphibia, Urodela), using indirect immunofluorescence techniques [167]. In *X. laevis*, NO activates eggs parthenogenesis [168] and is involved in the development of the mouth and neural crest [169].

### NO and Metamorphosis in Anura

The metamorphosis in amphibians consists of the transformation of the larva to a miniature adult replicate, with a change in the lifestyle from aquatic to terrestrial or semi-terrestrial [170]. Metamorphosis is guided by the hormone thyroxine, and it is nearly unnoticeable in Apoda and Urodela, but dramatic in Anura [170]. Anuran larvae (tadpoles) are strikingly different from the adult. Their metamorphosis requires comprehensive structural and physiological reorganization, involving substantial histological and physiological changes: absorption of the extensive tadpole tail and development of limbs and complex multicellular skin glands, rearrangement of the visual system, gut alterations as the animal changes from a herbivore to a carnivore, and gill resorption as gas exchange occurs in newly developed lungs as the amphibian adapts from an aquatic to a terrestrial environment [170].

This change in the anuran body plan is associated with profound changes in the behavioral-locomotor system, with a gradual switch from swimming, in which trust is generated by undulatory oscillations of the trunk and tail, to the limb-based propulsion of adults. This is associated with the degeneration of the Mauthner neurons (a pair of neural cells involved in controlling the escape response of tadpoles) and the rearrangement of the visual system from a lateral position (due to panoramic vision, an adaptation to the herbivorous prey lifestyle of tadpoles) to a frontal position that is appropriate for the predatory lifestyle of adults [162].

All these changes require critical but gradual modifications in the central and peripheral nervous pathways involved in controlling locomotory and visual functions [171] that are consistent with the behavioral requirements for continuous locomotor capability throughout metamorphosis.

NO appears to be deeply involved in the anuran metamorphosis process, particularly the dramatic change in locomotion modality and related behavioral patterns. The role of NO in reconfiguring the sensory and locomotor neural networks during metamorphosis has been proposed in *Rana esculenta* by identifying Nos immunoreactivity in the root ganglia and dorsal horn [172]. In *X. laevis*, the distribution of *Nos1* and *Nos2* was found to be stage-specific, and the *Nos1* gene expression was up-regulated by thyroxin treatment [173].

NO is one of the three neuromodulators, together with dopamine and serotonin, that potently influence CPG outputs in the development of *X. laevis* tadpole locomotion [162]. Their roles switch from predominantly inhibitory early in development to mainly excitatory at the later stages. Nos-positive neurons appear very early in the *Xenopus* brainstem. Until stage 42, NO exerts an inhibitory role on tadpole swimming via the presynaptic facilitation of inhibitory transmitter release [162]. At this stage, the tadpole lives on yolk and swims only when stimulated by potential predators, remaining otherwise motionless. Successively, Nos-positive neuron populations expand through the brain and spinal cord, and the NO role switches from inhibitory to facilitatory, although the mechanism involved is unclear [162]. This switch is associated with a dramatic change in locomotor behavior in *Xenopus* tadpoles as they become free-swimming and active filter feeders.

Another dramatic change in amphibian physiology during metamorphosis is in the respiratory system, from gill-based aquatic respiration to lung-based air respiration. NO is also involved in this process. Early in development, NO provides a tonic inhibitory input to gill and lung burst activity, but as development progresses, NO provides an excitatory input to lung ventilation [174]. Likely, NO provides a tonic input to the respiratory CPG during development, and this changing role reflects the modulatory influence of NO on inhibitory or excitatory modulators or neurotransmitters involved in the generation of respiratory rhythm [174].

## 8. Conservation of NO Functions in Aquatic Chordates

The analysis of NO’s multiple functions in aquatic chordates revealed a series of conserved roles both during embryonic development and in adult organism life. This gaseous molecule provides regulation and fine-tuning of a wide range of physiological processes (Figure 3).

A common NO function among different chordate species is in the maturation of eggs and sperm. Similarly to teleosts, and also in cephalochordates, NO is involved in the maturation of germinal cells [15,50,51,67,68,81]. An analogue function in tunicates could be suggested by the presence of NADPH-diaphorase in adult gonads (Figure 3) [75].

During development, NO plays crucial roles in the definition of head structures in all chordate species investigated so far. The Nos1 loss-of-function induces severe defects in the anterior structures in vertebrate *Xenopus* and zebrafish [169]. Similarly, endogenous NO reduction in amphioxus during development prevents the correct development of the mouth and pharyngeal structures [45,49]

In ascidians, the larval palps represent the most anterior sensory organs where Nos is expressed, further confirming the conservation of this messenger activity in the development of the anterior sensory regions among chordates (Figure 3). In tunicates, these anterior structures assure larval settlement. Interestingly, NO has a conserved function in the regulation of larval settlement in all metazoans undergoing metamorphosis [175]. As reported in other chordates, NO’s prominent role in the larval settlement is the modulation of the CPG neural circuits for the swimming behavior of tunicate larvae [5] and for fish locomotion [162,163]. As pointed out in Figure 3 (question mark), this role has not yet been investigated in cephalochordates. Nevertheless, since this role is present in the other chordates and in non-chordate invertebrates [176,177], it is most likely also conserved in amphioxus.

The ascidian larval settlement is followed by metamorphosis induction, where NO’s neuromodulatory role is also highly conserved (Figure 3). An interesting parallel can be observed comparing NO activities in tail retraction in chordates with an indirect development [5,46,178].

A relatively well-defined and conserved role of NO is the modulatory activity of the chordate immune system (Figure 3). In adult cephalochordates and tunicates, NO is present in immune-system-associated organs and cells [50,51,67,76,77]. Accordingly, Nos2 activation in the immune response to infection is observed in various teleost species [139,140].

In the adult stage of aquatic vertebrates, NO is likely involved in all the main physiological processes, such as sensory information processing, cardiorespiratory function, feeding and digestion, neuroendocrine regulation, and behavior [156]. It is not easy to detail the parallelism among tunicates, teleosts, and amphibians. However, the data on tunicate NADPH localization in the adult endostyle and in the nerve filaments innervating the muscles of the heart, the digestive system, and the gill sac suggest the occurrence of these NO functions already in the invertebrate chordates (Figure 3) [75]. These roles have considerably specialized following the development of the more complex lifestyles of vertebrates.

In conclusion, our comparative analyses of *Nos* expression and NO function in aquatic chordates evidenced a high degree of conservation from non-vertebrate chordates to vertebrates from development to the adult stage. Nevertheless, further functional studies will be necessary for non-vertebrate chordates to confirm this hypothesis, sometimes inferred only from common data on expression or behavior.

## Figures and Tables

**Figure 1 ijms-24-11182-f001:**
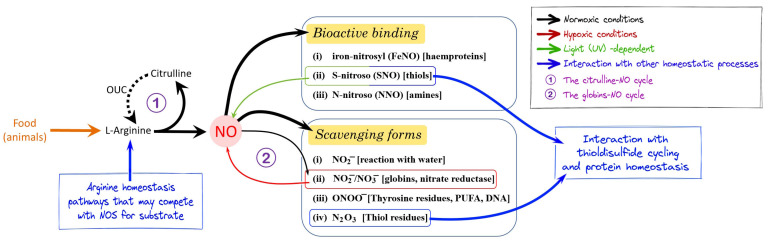
Schematic resuming of all major transformations putatively involved in nitric oxide (NO) homeostasis. Under normoxic conditions, NO is synthesized by Nos enzymes from L-arginine. In organisms possessing the ornithine–urea cycle (OUC), the co-product of the reaction, L-citrulline, is reconverted to L-arginine (the citrulline–NO cycle). NO signaling may occur by (**i**) forming an iron-nitrosyl complex with hemoproteins; (**ii**) promoting protein post-translational modifications via S-nitrosylation, forming S-nitroso compounds with thiol residues (SNO); and (**iii**) forming N-nitroso (NNO) compounds. NO can be (**i**) rapidly converted to nitrite with dissolved O_2_ by NO scavenging reactions or can be (**ii**) inactivated to nitrate by oxygenated heme proteins. Excess NO (**iii**) can react with superoxide to form peroxynitrite (ONOO^−^), which in turn can interact with lipids, DNA, and proteins to give nitrated derivatives or (**iv**) react with oxygen to form N_2_O_3_, which interacts with thiol groups to give nitrosylated proteins. When *de novo* production via Nos is compromised under low O_2_, NO can be produced by nitrite reduction in heme proteins (the globin–NO cycle), possibly associated with nitrate conversion to nitrite by xanthine oxidoreductase. The interactions with arginine and protein homeostasis may be significant in determining the balance between NO formation and scavenging.

**Figure 2 ijms-24-11182-f002:**
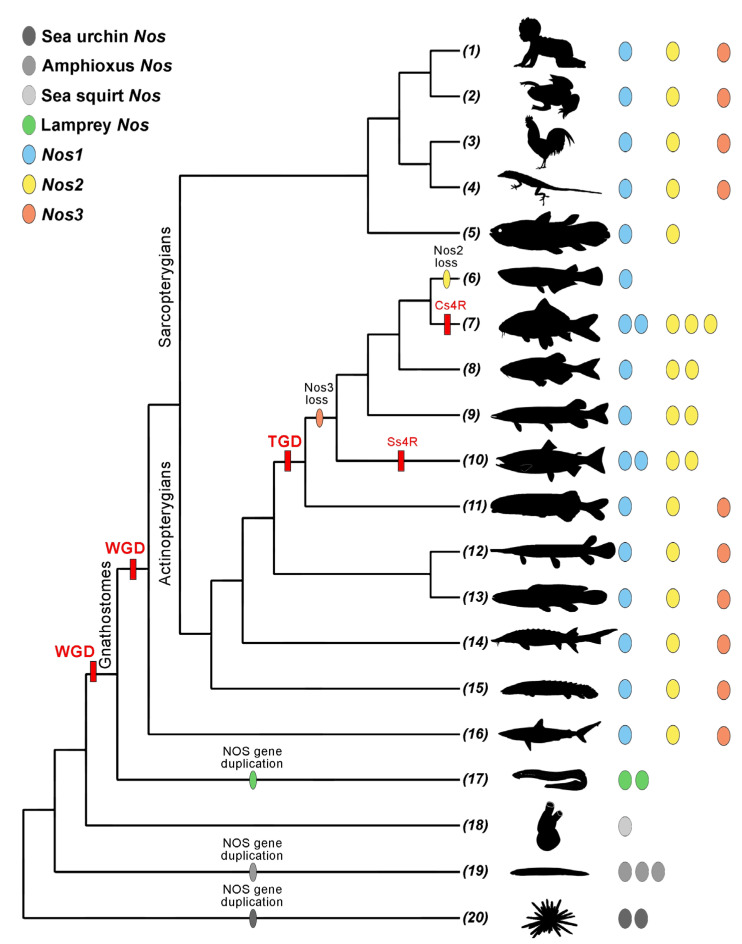
*Nos* repertoire in deuterostomes. Silhouettes are from PhyloPic (PhyloPic—free silhouette images of life forms) and represent the following: (1) Human (*Homo sapiens*); (2) Frog (*Xenopus laevis* and *Xenopus tropicalis*); (3) Chicken (*Gallus gallus*); (4) Lizard (*Anolis carolinensis*); (5) Coelacanth (*Latimeria chalumnae*); (6) Medaka (*Oryzias latipes*); (7) Carp (*Carassius auratus, Cyprinus carpio,* and *Sinocyclocheilus anshuiensis*); (8) Zebrafish (*Danio rerio*); (9) Northern pike (*Esox lucius*); (10) Salmon (*Oncorhynchus mykiss* and *Salmo salar*); (11) Elephant fish (*Paramormyrops kingsleyae*); (12) Spotted gar (*Lepisosteus oculatus*); (13) Bowfin (*Amia calva*); (14) Sturgeon (*Acipenser ruthenus*); (15) Bichir (*Polypterus senegalus*); (16) Shark (*Scyliorhinus torazame*); (17) Lamprey (*Lethenteron camtschaticum*, *Petromyzon marinus*); (18) Sea squirt (*Ciona robusta* and *Ciona savignyi*); (19) Amphioxus (*Asymmetron lucayanum, Branchiostoma belcheri, Branchiostoma floridae,* and *Branchiostoma lanceolatum*); (20) Sea urchin (*Hemicentrotus pulcherrimus* and *Strongylocentrotus purpuratus*). Abbreviations: Cs4R, carp-specific genome duplication; Ss4R, salmonid-specific genome duplication; TGD, teleost genome duplication; VGD, vertebrate genome duplication.

**Figure 3 ijms-24-11182-f003:**
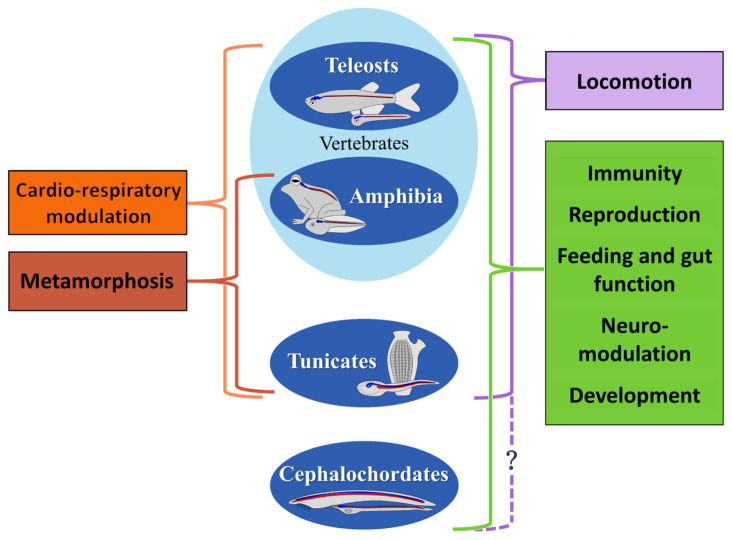
Comparison of the major physiological functions of NO in aquatic chordates. The NO roles in immunity, reproduction, feeding and gut function, neuro-modulation, and development are common to all chordates (green). The role in locomotion is conserved among vertebrates and tunicates (lilac), but no data are available in cephalochordates (question mark). NO function in metamorphosis is common to all chordates with an indirect development (light brown). Cardio-respiratory modulation by NO is reported in vertebrates and tunicates (orange).

**Table 1 ijms-24-11182-t001:** The oxidized or reduced derivatives of NO in biological systems, forming the complex of NO_x_ (i.e., the pool of NO and its oxidized or reduced derivative species that can be found in living cells) [27,28].

Type	Species	Formula	Role
oxidized	nitrate	NO_3_^−^	intermediate of NO transformation and source of nitrite under hypoxia
oxidized	nitrite	NO_2_^−^	signaling agent; intermediate of NO transformation
oxidized	peroxynitrite	ONOO^−^	intermediate of NO transformation; by-product under oxidative stress
oxidized	nitrogen dioxide	NO_2_	intermediate of NO transformation; by-product under oxidative stress
oxidized	dinitrogen dioxide	N_2_O_3_	mobile NO transporter
reduced	nitroxyl/nitroxide ion	HNO/NO^−^	signaling agent
reduced	nitroso-thiols	RS-NO	signaling agent; mobile NO transporter
reduced	hyponitrite	N_2_O_2_^2−^	intermediate of NO transformation

**Table 2 ijms-24-11182-t002:** The table reports the multiple acronyms of *Nos* genes and proteins present in the literature. The selected nomenclature for duplicate Nos takes into account the gene or genome duplication events. * Nomenclature adopted in this manuscript.

Extended Nomenclature	Different Acronyms Present in Literature	Duplicates Nomenclature
neuronal Nos	nNos, nNOS, bNos, NOSI, NOS1, Nos1 *	Nos1a–Nos1b
		Nos1α–Nos1β
inducible Nos	iNos, iNOS, NOSII, NOS2, Nos2 *	Nos2a–Nos2b
		Nos2ba–Nos2bb
		Nos2.1–Nos2.2
		Nos2α–Nos2β
endothelial Nos	eNos, eNOS, NOSIII, NOS3, Nos3 *	

**Table 3 ijms-24-11182-t003:** NOS gene repertoire in deuterostomes. Table reports exhaustive information on *Nos* genes available in deuterostomes, with a special focus on chordates. The number of Nos orthologs per each species and the corresponding bibliographic references are shown. The symbol # indicates that *Ciona robusta* was previously named *Ciona intestinalis*, and * indicates that a third *Nos* gene should be expected in those species, although one of the orthologs is still missing, probably for genomic quality issues.

		DEUTEROSTOMES				*Nos* GENES	REFs
AMBULACRARIA	Echinoderms	Echinoids		*Strongylocentrotus purpuratus* *Hemicentrotus pulcherrimus*		2	[42] [47] [48]
CHORDATES	Cephalochordates			*Asymmetron lucayanum*		3	[42] [45] [49] [50] [51]
				*Branchiostoma floridae* *Branchiostoma lanceolatum* *Branchiostoma belcheri*		3	
	Urochordates	Ascidians		*Ciona robusta* ^#^ *Ciona savignyi*		1	[42] [46] [52]
		Cyclostomes		*Lethenteron camtschaticum* *Petromyzon marinus*		2	[42] [53] [54] [55]
				*Eptatretus burgeri*			
				Chondrichthyes	*Scyliorhinus torazame*	3	
					*Chiloscyllium punctatum*	2 *	
					*Rhincodon typus*	2 *	
					*Callorhinchus milii*	2 *	
		Actinopterygians	Polypteriformes	*Polypterus senegalus*		3	
			Acipenseriformes	*Acipenser ruthenus*		3	
				Holosteans	*Amia calva* *Lepisosteus oculatus*	3	
				Teleosts	*Paramormyrops kingsleyae*	3	
					*Salmo salar*	4	
					*Esox lucius*	3	
					*Danio rerio*	3	
					*Cyprinus carpio*	5	
					*Oryzias latipes*	1	
		Sarcopterygians	Coelacathiformes		*Latimeria chalumnae*	2 *	
				Tetrapods	*Homo sapiens* *Anolis carolinensis* *Gallus gallus* *Xenopus tropicalis*	3	

## Data Availability

All data is available upon request.

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
