# Peer review of "Nitric Oxide Function and Nitric Oxide Synthase Evolution in Aquatic Chordates"

_ijms, 2023, doi:10.3390/ijms241311182_

Round 1

Reviewer 1 Report

The review entitled “Nitric oxide function and Nitric Oxide Synthase evolution in aquatic chordates” from Locascio and colleagues, provides a valuable and interesting comparative analysis on the function of NO in the aquatic chordates tunicates, cephalochordates, teleost fishes and amphibians. The paper is well written and it can be accepted for publication after minor revision.

Specific Comments

Figure 1. I suggest to use the same font unless different fonts are necessary to better describe pathways.

line 357: please, also add reference [77] since it contains a more detailed description of species in which an eNos-like isoform has been found.

lines 354-364: description of Nos isoforms and the role of NO in the cardiovascular system appears confused moving back and forth between vessels and the heart. Please, rephrase it also taking into consideration that Nos1 signals have been detected in intracardiac neurons and axons in the sinoatrial plexus of the goldfish heart (Newton et al. 2014; J Comp Neurol 522:456–478).

8. Conservation of NO functions in aquatic chordates. This paragraph may be reduced in content since some concepts have been already described.

Author Response

The review entitled “Nitric oxide function and Nitric Oxide Synthase evolution in aquatic chordates” from Locascio and colleagues, provides a valuable and interesting comparative analysis on the function of NO in the aquatic chordates tunicates, cephalochordates, teleost fishes and amphibians. The paper is well written and it can be accepted for publication after minor revision.

Specific Comments

Figure 1. I suggest to use the same font unless different fonts are necessary to better describe pathways.

We prefer to use different fonts to better highlight all major transformations of NO and their relevance to homeostasis.

line 357: please, also add reference [77] since it contains a more detailed description of species in which an eNos-like isoform has been found.

Reference 77 (now 80) was added.

lines 354-364: description of Nos isoforms and the role of NO in the cardiovascular system appears confused moving back and forth between vessels and the heart. Please, rephrase it also taking into consideration that Nos1 signals have been detected in intracardiac neurons and axons in the sinoatrial plexus of the goldfish heart (Newton et al. 2014; J Comp Neurol 522:456–478).

The text was rewritten and the suggested reference was added.

  1. Conservation of NO functions in aquatic chordates. This paragraph may be reduced in content since some concepts have been already described.

The paragraph was reduced accordingly to the advices of the referee.

Reviewer 2 Report

The manuscript by Anamaria Locascio et al. entitled: “Nitric oxide function and Nitric Oxide Synthase evolution in aquatic chordates” reviews the conserved functions of nitric oxide in aquatic chordates, particularly during development, and also provides information on the evolution of nitric oxide synthase encoding genes. The review is divided in several sections. The first presents NO homeostasis, then NOS evolution in deuterostomes (i.e. the superphylum that includes three clades: Chordata (e.g. all vertebrates), Echinodermata (e.g. starfish), and Hemichordata. Then the NO functions in several groups of fish and amphibians, with a focus on the role of NO in larva-adult metamorphosis in the order Anura, are presented and the review ends with a presentation of the conservation of NO functions in aquatic chordates. The bibliography presents 171 references. I found reading this comprehensive review very interesting and the structure and focus of this manuscript makes it suitable as a review article. It is recommended that this paper be published after the following minor issues have been addressed by the authors.

1. A new Table should be included that would define explicitly the correspondence between Nps1, Nos2 introduced in this manuscript by the authors as a new nomenclature for nitric oxide synthases, and the classical names: bNOS, eNos, iNos and nNOS (names that are still very commonly used in the literature) to help the reader to better follow what the authors expose.

2. I strongly suggest that the authors add to their Fig. 2 which species illustrated belong to cephalochordates (reviewed in Section 4), to tunicates (reviewed in Section 5), to teleosts (reviewed in Section 6) and to amphibians (reviewed in Section 7). This would greatky help the general reader.

3. Lines 64-65. The sentence “NO bind ferrous heme groups with a five-coordinated geometry, unlike O2 and CO that employ a six-coordinate geometry” as such is not correct. Instead of five- or six-coordinate “geometry”, they should write: five- or six-coordinate “species”. They could also refer to low-spin or high-spin heme species, which is more correct physiochemically speaking. Moreover, the paper in Ref. 3 cited for this sentence provide much more information on NO binding to metal ion that is not presented in this manuscript. NO can bind to non-heme iron, such as iron ions in FeS clusters forming nitrosyl iron species. No can also bind to copper ion, reducing Cu2+ to Cu+ in mitochondrial cytochrome c oxidase. These aspects are not resented int his review, though they can have profound consequences on cell physiology.

4. Lines 113-114. This is the only occurrence where H2S is cited and put in relation with NO functions. I find that this review is not sufficiently putting in perspectives what could be learnt from aquatic chordates evolution and the putative role(s) of NO and H2S in the primitive (prebiotic) atmosphere. Could the authors comment on that?

5. Question. Why is it that coelacanth (species 5 in Fig. 2) has only two NOSs though belonging to the sarcopterygians branch where all other organisms have three NOSs? Could the authors comment on that?

6. Lines 188-189. The sentence here reads: “The loss of Nos3 has been detected in all fish lineages analyzed…” Nos3 is represented as the orange ovale, and this orange item is seen in several fish lineages such as those from 11 to 16, making this sentence at lines 188-189 not correct. Could the authors correct this?

7. Minor. In Fig. 2 legend, abbreviations Ss4R and Cs4R are not defined.

8. Minor. Line 48. The wrong word “chemical-physical” should be changed to “physicochemical” or to “chemicophysical”, either.

9. Minor. Line 384. The abbreviation “SW” is not defined.

10. Minor. Line 465. The abbreviation “19 hpf” is not defined.

11. In References, there are redundant information for some references such as Ref.3 where “2022, 11, 122” is written twice. There are other references with the same redundancy.

12. Last but not least. Lien 449. The authors present the function of NO in “the expression of social behaviors”. They should add this to the list in Fig. 3. In this respect, the authors could maybe present briefly the role of NO in social fear contagion in zebrafish via the oxytocin system. The role of oxytocin in regulating social fear contagion has been very recently brilliantly demonstrated (See: Akinrinade et al. Science 23 mar 2023, 379(6638), 1232-1237) and the role of No in oxytocin function has been known since a long time (See: Rettori et al. Braz J Med Biol Res 1997 Apr, 30(4), 453-457).

Author Response

The manuscript by Anamaria Locascio et al. entitled: “Nitric oxide function and Nitric Oxide Synthase evolution in aquatic chordates” reviews the conserved functions of nitric oxide in aquatic chordates, particularly during development, and also provides information on the evolution of nitric oxide synthase encoding genes. The review is divided in several sections. The first presents NO homeostasis, then NOS evolution in deuterostomes i.e. the superphylum that includes three clades: Chordata (e.g. all vertebrates), Echinodermata (e.g. starfish), and Hemichordata. Then the NO functions in several groups of fish and amphibians, with a focus on the role of NO in larva-adult metamorphosis in the order Anura, are presented and the review ends with a presentation of the conservation of NO functions in aquatic chordates. The bibliography presents 171 references. I found reading this comprehensive review very interesting and the structure and focus of this manuscript makes it suitable as a review article. It is recommended that this paper be published after the following minor issues have been addressed by the authors.

  1. A new Table should be included that would define explicitly the correspondence between Nos1, Nos2 introduced in this manuscript by the authors as a new nomenclature for nitric oxide synthases, and the classical names: bNOS, eNos, iNos and nNOS (names that are still very commonly used in the literature) to help the reader to better follow what the authors expose.

We included a new Table (Table 2) to clarify the nomenclature choice we made and all gene names reported in literature.

  1. I strongly suggest that the authors add to their Fig. 2 which species illustrated belong to cephalochordates (reviewed in Section 4), to tunicates (reviewed in Section 5), to teleosts (reviewed in Section 6) and to amphibians (reviewed in Section 7). This would greatky help the general reader.

We included in the legend of Figure 2 the name of the species used in the different sections of this article.

  1. Lines 64-65. The sentence “NO bind ferrous heme groups with a five-coordinated geometry, unlike O2 and CO that employ a six-coordinate geometry” as such is not correct. Instead of five- or six-coordinate “geometry”, they should write: five- or six-coordinate “species”. They could also refer to low-spin or high-spin heme species, which is more correct physiochemically speaking. Moreover, the paper in Ref. 3 cited for this sentence provide much more information on NO binding to metal ion that is not presented in this manuscript. NO can bind to non-heme iron, such as iron ions in FeS clusters forming nitrosyl iron species. No can also bind to copper ion, reducing Cu2+ to Cu+ in mitochondrial cytochrome c oxidase. These aspects are not presented in this review, though they can have profound consequences on cell physiology.

We have modified the text accordingly to the referee’s suggestion. We have also provided information on NO binding to non-heme iron and copper ions.

Lines 113-114. This is the only occurrence where H2S is cited and put in relation with NO functions. I find that this review is not sufficiently putting in perspectives what could be learnt from aquatic chordates evolution and the putative role(s) of NO and H2S in the primitive (prebiotic) atmosphere. Could the authors comment on that?

Our aim was to mention that Nos activity can be modulated by H2S. The putative roles of NOS and H2S in the primitive prebiotic atmosphere is outside the scope of the paper. Other reviews address this aspect in details.

  1. Question. Why is it that coelacanth (species 5 in Fig. 2) has only two NOSs though belonging to the sarcopterygians branch where all other organisms have three NOSs? Could the authors comment on that?

The reviewer’s observation is correct. We tried to find a bona fide Nos3 in coelacanth Latimeria chalumnae genome, but failed to retrieve a reliable sequence. This means that either coelacanths lost the Nos3 gene during their evolution or this information is missing in the only available genome assembly due to sequencing gaps. This was already mentioned in the legend of (now) Table 3. Hopefully, in the next future, sequencing projects of the other known coelacanth Latimeria menadoensis could help to clarify this issue.

  1. Lines 188-189. The sentence here reads: “The loss of Nos3 has been detected in all fish lineages analyzed…” Nos3 is represented as the orange ovale, and this orange item is seen in several fish lineages such as those from 11 to 16, making this sentence at lines 188-189 not correct. Could the authors correct this?

We simplified the figure avoiding too many details of duplications and eliminated the gene duplication event in lineage (9). Regarding the text we substituted, at lines 196, the word “fish” with “teleosts” (lineages 6-7-8-9-10).

  1. Minor. In Fig. 2 legend, abbreviations Ss4R and Cs4R are not defined.

We have defined the abbreviations Ss4R and Cs4R in Fig, 2 legend.

  1. Minor. Line 48. The wrong word “chemical-physical” should be changed to “physicochemical” or to “chemicophysical”, either.

We have reported the corrected word.

  1. Minor. Line 384. The abbreviation “SW” is not defined.

We have removed “SW” abbreviation and reported “sea water”

  1. Minor. Line 465. The abbreviation “19 hpf” is not defined.

We have defined the “19hpf” abbreviation

  1. In References, there are redundant information for some references such as Ref.3 where “2022, 11, 122” is written twice. There are other references with the same redundancy.

All references were checked

  1. Last but not least. Lien 449. The authors present the function of NO in “the expression of social behaviors”. They should add this to the list in Fig. 3. In this respect, the authors could maybe present briefly the role of NO in social fear contagion in zebrafish via the oxytocin system. The role of oxytocin in regulating social fear contagion has been very recently brilliantly demonstrated (See: Akinrinade et al. Science 23 mar 2023, 379(6638), 1232-1237) and the role of No in oxytocin function has been known since a long time (See: Rettori et al. Braz J Med Biol Res 1997 Apr, 30(4), 453-457).

As suggested, we have added in the text the role of NO in social fear contagion in zebrafish via the oxytocin system. We have not enough data to include “Social behaviors” among the common functions of NO in different aquatic chordate species.